# Efficacy and Safety of Anticoagulant Therapy in COVID-19-Related Pulmonary Embolism with Different Extension

**DOI:** 10.3390/biomedicines11051282

**Published:** 2023-04-26

**Authors:** Maria Chiara Gatto, Alessandra Oliva, Claudia Palazzolo, Claudio Picariello, Andrea Garascia, Emanuele Nicastri, Enrico Girardi, Andrea Antinori

**Affiliations:** 1National Institute for Infectious Diseases, Lazzaro Spallanzani, IRCCS, Via Portuense, 292, 00149 Rome, Italy; 2Department of Public Health and Infectious Disease, Sapienza University of Rome, Piazzale Aldo Moro n.5, 00185 Rome, Italy; 3UOC Cardiologia, Azienda Ospedaliera Santa Maria della Misericordia, ULSS5 Polesana, 45100 Rovigo, Italy; 4Department of Cardiology, De Gasperis Cardio Center, ASST Grande Ospedale Metropolitano Niguarda, 20162 Milan, Italy

**Keywords:** SARS-CoV-2, COVID-19, pulmonary embolism, anticoagulation therapy

## Abstract

Pulmonary embolism (PE) has been associated with SARS-CoV-2 infection, and its incidence is highly variable. The aim of our study was to describe the radiological and clinical presentations, as well as the therapeutic management, of PEs that occurred during SARS-CoV-2 infection in a cohort of hospitalized patients. In this observational study, we enrolled patients with moderate COVID-19 who developed PE during hospitalization. Clinical, laboratory, and radiological features were recorded. PE was diagnosed on clinical suspicion and/or CT angiography. According to CT angiography results, two groups of patients were further distinguished: those with proximal or central pulmonary embolism (cPE) and those with distal or micro-pulmonary embolism (mPE). A total of 56 patients with a mean age of 78 ± 15 years were included. Overall, PE occurred after a median of 2 days from hospitalization (range 0–47 days) and, interestingly, the majority of them (89%) within the first 10 days of hospitalization, without differences between the groups. Patients with cPE were younger (*p* = 0.02), with a lower creatinine clearance (*p* = 0.04), and tended to have a higher body weight (*p* = 0.059) and higher D-Dimer values (*p* = 0.059) than patients with mPE. In all patients, low-weight molecular heparin (LWMH) at anticoagulant dosage was promptly started as soon as PE was diagnosed. After a mean of 16 ± 9 days, 94% of patients with cPE were switched to oral anticoagulant (OAC) therapy, which was a direct oral anticoagulant (DOAC) in 86% of cases. In contrast, only in 68% of patients with mPE, the prosecution with OAC was indicated. The duration of treatment was at least 3 months from PE diagnosis in all patients who started OAC. At the 3-month follow-up, no persistence or recurrence of PE as well as no clinically relevant bleedings were found in both groups. In conclusion, pulmonary embolism in patients with SARS-CoV-2 may have different extensions. Used with clinical judgment, oral anticoagulant therapy with DOAC was effective and safe.

## 1. Introduction

Venous and arterial thrombosis have been widely described during SARS-CoV-2 infection. Different pathogenetic mechanisms, including, but not limited to, in situ immune-thrombosis, sNox-2 activation, or embolisms from a distant site could explain these phenomena [1,2,3]. In particular, the virus interacts with the type II pneumocytes through the membrane-bound angiotensin-converting enzyme 2. This interaction can lead to a pneumocyte activation, production of thrombo-inflammatory cytokines, platelet activation, NETs formation, and coagulation stimulation, promoting an immune pulmonary intravascular coagulopathy, which results in an in situ thrombosis [4].

Since the beginning of the pandemic, pulmonary embolism (PE) has been associated with SARS-CoV-2 infection, and an increased number of hospitalizations for PE was recorded [5,6]. As a matter of fact, registry data showed a tenfold increment of hospitalization for PE at the Lazzaro Spallanzani National Institute for Infectious Diseases in Rome [7]. However, the incidence of PE in patients with SARS-CoV-2 infection is highly variable among studies, ranging from 2–15% to 17–70% in patients admitted to the intensive care unit (ICU) or general wards, respectively [8,9,10,11]. Indeed, a recent metanalysis involving nearly 34,000 patients showed that, when systematically researched, PE has an incidence of up to 40% [12]. Likewise, in patients hospitalized for COVID-19, clinical, laboratory, and instrumental presentations of PE are highly variable [13,14].

Predictors of PE in these patients are, among others, ICU admission, male sex, and very high D-Dimer levels [1,2,3]. While diagnostic algorithms based on pre-test probability assessment such as the Padua score (Appendix A) are widely used for predicting the risk of PE also before the COVID-19 pandemic [15], the ADA (Age, D-Dimer, Albumin) score [16] has been recently validated in COVID-19 patients. However, the gold standard for the diagnosis of PE is the chest CT Angiography [17,18].

Anticoagulant therapies with low-molecular-weight heparin (LMWH) or direct oral anticoagulants (DOAC) are the mainstay of therapeutic management of PE. However, whether to choose LMWH or DOAC should be carefully considered, and the duration of treatment in COVID-19 patients may depend also on the clinical presentation and radiological extension of PE [19,20,21,22,23,24]. Moreover, in the first months of the ongoing pandemic, the antimicrobial therapy, characterized by the use of hydroxychloroquine, lopinavir, and azytromicin, had potential and serious interactions with DOACs. At present, one of the main antiviral drugs used in the early stages of COVID-19 (nirmatrelvir/ritonavir) may have interactions with DOACs [3].

This study aims to describe the features, the diagnostic process, the therapeutic indications, and the three-month follow-up of prospectively enrolled adult patients hospitalized for SARS-CoV-2 infection and developing PE.

## 2. Materials and Methods

A prospective observational study of patients hospitalized with COVID-19 and diagnosed with PE has been conducted. The patients were admitted to the medical units of Lazzaro Spallanzani National Institute of Infectious Diseases in Rome between February and August 2022. All available data about clinical features, laboratory parameters, and radiological characteristics were collected; laboratory data were recorded at the diagnosis (±24 h) of PE. SARS-CoV-2 infection was confirmed by means of a rapid antigen or molecular (Real-Time PCR) nasopharyngeal swab test, when indicated. All included patients had been hospitalized for moderate SARS-CoV-2 infection defined as “Evidence of lower respiratory disease during clinical assessment or imaging, with SpO2 ≥ 94% on room air at sea level”, according to the Coronavirus Disease 2019 Treatment Guidelines released by the National Institute of Health [25]. The characteristics of the population were described, and some scores were calculated. The CHA_2_DS_2_-VASc score was calculated as it is correlated with mortality regardless of the presence or absence of atrial fibrillation [26,27].

The diagnosis of PE was obtained through CT angiography according to the latest guidelines [17,21], in the case of unexpected worsening of respiratory function, new-onset tachycardia, abrupt reduction in blood pressure, in the case of electrocardiographic changes (e.g., S1-Q3-T3 pattern and/or negative T waves in the inferior site and/or new-onset right bundle branch block), or in the case of deep-vein thrombosis (DVT). Furthermore, if clinical presentation was strongly suggestive (Appendix A) of PE despite a D-Dimer in the normal range, pulmonary CT angiography was performed anyway, in accordance with current guidelines [21].

Based on CT angiography findings, we defined micro-pulmonary embolism (mPE), characterized by distal and partial filling defects in the segmental and/or subsegmental branches, and proximal or central pulmonary embolism (cPE) (Figure 1a) [28], characterized by the involvement of principal pulmonary arteries (Figure 1b). In one case, PE was associated with left ventricular thrombosis (Figure 1c).

In accordance with the current guidelines [21], our practical management of PE in patients with SARS-CoV-2 infection was based on the so-called “double drug approach”, consisting in an initial therapy with LMWH (fondaparinux or enoxaparin at anticoagulant dosage) for at least 5–7 days, followed by oral anticoagulation, which was prosecuted after hospital discharge. The decision to use LMWH first was based on its anti-inflammatory properties [19,22] and the absence of drug–drug interactions with some commonly used antivirals (nirmatrelvir/ritonavir) that, in contrast, DOACs may have [3]. Among the currently available DOACs, we generally prefer Dabigatran or Edoxaban for their easier maneuverability in the acute phase of infection and the possibility of dose reduction as indicated in the drug’s technical data sheet (Appendix A).

Follow-up visits at day 30 and month 3 were performed in all patients enrolled.

The Strengthening the Reporting of Observational studies in Epidemiology (STROBE) guidelines were applied to ensure high-quality presentation of the conducted observational study (Appendix A).

### Statistical Analyses

Continuous variables were reported as mean and standard deviation (SD) or median and 25–75th Interquartile Range (IQR), when indicated, while categorical variables were reported as absolute number and percentage. Differences between groups were evaluated with Student’s *t*-test or Mann–Whitney test; the difference between proportions was calculated by Chi-square or Fisher’s exact tests, as appropriate. The time to PE development was analysed by Kaplan–Meier curves, and statistical significance of the differences between the groups (mPE vs. cPE) was assessed using the log-rank test. The data were recorded in anonymized spreadsheets, and the statistical analysis was blindly performed with the Prism GraphPad (v.8 produced by Dotmatics, Boston, MA, USA) or STATA (v.17 produced by StataCorp LLC, College Station, TX, USA) software, when appropriate.

## 3. Results

A total of 56 patients with a mean age of 78 ± 15 years were included, and 36 (72%) were men. The mean BMI was 26.1 ± 4.71 kg/m^2^, without a significant difference between male and female. The mean weight was 76 ± 19 kg, higher for men (81 ± 20 kg) than for women (65 ± 10 kg) (*p* = 0.003).

Overall, PE occurred after a median of 2 days from hospitalization (mean 6 ± 10, range 0–47 days) and, interestingly, the majority of them (89%) within the first 10 days of hospitalization (Figure 2a). Thirty-one patients (55%) were diagnosed with cPE and 25 patients (45%) with mPE. Time to diagnosis did not differ between cPE and mPE (log-rank 0.468) (Figure 2b), as well as the length of hospitalization: 25 ± 11 versus 30 ± 18 days, respectively (*p* = 0.678). Only in one case a concomitant left ventricular thrombosis (Figure 1c).

The clinical and laboratory features of the population are summarized in Table 1. Charlson Comorbidity Index (CCI), CHA_2_DS_2_-VASc, and HAS-BLED scores were recorded for all patients, and no difference was found between cPE and mPE. Interestingly, the presence of cancer (solid tumours or leukaemia) was more frequent in cPE patients, while coronary artery disease (CAD) was mostly observed in mPE.

Patients with cPE were younger (76 ± 13 vs. 81 ± 16 years, *p* = 0.02), with a lower creatinine clearance (66 ± 34 vs. 82 ± 38 mL/min, *p* = 0.04), and tended to have a higher body weight (81 ± 23 vs. 70 ± 12 kg, *p* = 0.059) and D-Dimer values [median 1735 (IQR 1155–5175) ng/mL versus 1380 (IQR 500–3600) ng/mL, *p* = 0.059] than patients with mPE.

In all patients, LWMH was promptly started at full therapeutic dose as soon as PE was diagnosed. In 29 patients (94%) with cPE, therapy with LWMH was switched to oral anticoagulation therapy after a mean of 15 ± 6 days, while the LWMH-to-OAC switch occurred only in 17 patients (68%) with mPE (*p* < 0.001). The patients who were not switched to OACs continued LWMH for 10 ± 5 days after discharge.

As oral anticoagulant therapy, DOACs were used in 25 (86%) and 14 (82%) patients with cPE and mPE, respectively. In these patients, a simpler “double drug approach” with Dabigatran or Edoxaban, which did not require the initial loading dose of OAC and whose dosage could be reduced according to the criteria indicated in the drug’s technical data sheet, was used in 27 cases (70%), while in the other 12 patients (30%), Apixaban and Rivaroxaban were used [3,21,24,29,30].

The duration of treatment was at least 3 months from PE diagnosis in all patients who started OAC therapy. In patients with a high risk of PE recurrence (oncological diseases, obesity, bed rest/hypomobility), anticoagulant therapy was continued beyond 3 months.

At the 30-day follow-up, four patients deceased due to severe COVID-19-related Acute Respiratory Distress Syndrome (ARDS) (2 in each group), whereas at the 3-month follow-up, no persistence or recurrence of PE as well as no clinically relevant bleedings were found in both groups. In Table 2, the management of anticoagulant therapy and follow-up outcomes are summarized.

## 4. Discussion

In this study, we analyzed patients with SARS-CoV-2 infection and different radiological findings of PE consisting in the involvement of central or peripherical branches of pulmonary circulation: cPE and mPE, respectively. Although several manifestations of venous and arterial thrombo-embolism have been described in the literature [1,2,3,5,6,8,9], to the best of our knowledge, no similar studies differentiating cPE from mPE have been conducted in COVID-19 patients. Furthermore, no studies have been published on the management of anticoagulant therapy specifically in patients with mPE.

The time of diagnosis of PE may vary depending on the clinical setting. In the literature, a median of 8 days is reported for outpatients [31]. In our study, which only considered hospitalized patients, the median time elapsed from COVID-19 diagnosis to PE was shorter (2 days), and the majority of PE cases (89%) occurred during the first 10 days from hospitalization, suggesting that utmost attention should be paid in the early phase of acute SARS-CoV-2 infection.

Actually, in the early stages of COVID-19, the widely used antiviral therapy with nirmatrelvir/ritonavir may have interactions with DOACs. Also for this reason, the use of heparin during treatment with these antivirals, which generally lasts no longer than five days, should be preferred [3]. Moreover, it should be noted that the anti-inflammatory effect of heparin during SARS-CoV-2 infection has been extensively described [32]. For this reason, it is rational to use LWMH for the initial treatment of PE in COVID-19, and it is equally reasonable to switch to oral DOAC therapy as soon as possible, especially if the patient is discharged.

The challenge for physicians is represented by the decision of whether, and how, to start anticoagulant therapy [15,18,21,22], especially in patients with mPE, that could be attributed to immune-thrombosis phenomena rather than embolism. Indeed, mPE has often been attributed to in situ and NET-osis in a context in which immune-mediated inflammation plays a key role [23,33]. In particular, during viral infections including also SARS-CoV-2 infection, the coagulation factors VIIa and Xa are released by monocytes and subendothelial cells; the endothelial cells and platelets release pro-inflammatory cytokines and pro-coagulant particles which, in turn, increase the recall of leukocytes. Neutrophil cells release an extracellular trap that amplifies the effects described above. The phenomenon, called “NET-osis”, causes endothelial and microvascular damage and consequent microthrombosis in situ or other, more extensive thrombotic phenomena [23].

Similarly, the inflammation plays a crucial role in vascular atherosclerosis and ischaemic heart disease [34]. With this regard, we found that patients with CAD developed mPE more frequently than cPE. On the other hand, patients with a cancer history developed more frequently cPE than mPE [35,36,37]. Therefore, in this context it may be difficult to distinguish whether PE, and in particular mPE, is due to the SARS-CoV-2 infection itself (i.e., PE “due to” COVID-19) or whether it is due to pre-existing causes (i.e., PE “with” COVID-19), such as cancer, obesity, bed rest, cardiovascular disease [35,36,37].

In our cohort, post-discharge anticoagulation with OAC was indicated for almost all patients with cPE (94%), while for patients with mPE it was used in only 68% of cases. This significant difference was probably related to the fact that mPE was considered as a non clinically relevant phenomenon occurring during the acute inflammation phase of SARS-CoV-2 infection.

Unfortunately, the long-term consequences of mPE are still unknown, as well as the role of chronic inflammation following the acute phase of COVID-19 [38,39]. During SARS-CoV-2 infection and follow-up, it is also important to assess D-Dimer as a marker of prognosis [40].

The interesting finding of the present study is that, in all patients on OAC therapy after discharge, no clinical relevant bleedings and no recurrence of PE were recorded at the three-month follow-up, suggesting that post-discharge anticoagulation therapy also in patients with mPE who have no contraindications or increased risk of bleeding may be considered a reasonable and safe choice. It is probably that, in mPE, anticoagulant treatment may be prescribed for a shorter period of time.

Undoubtedly, the study presents several limitations, such as the small number of subjects recruited and its single-center, observational nature. There are no data on compliance with anticoagulant therapy after discharge, and this may be a further limitation of the study. Despite such limitations, presenting these data may be of interest. Indeed, to the best of our knowledge, there are no studies in the literature comparing the different types of PE during SARS-CoV-2 infection. Moreover, there is no unanimous consensus on anticoagulant therapy in the distal forms of PE during COVID-19, especially with regard to long-term treatment. If SARS-CoV-2 infection is considered a predisposing and triggering factor for PE, according to the recent guidelines, anticoagulant therapy should be continued for at least three months, but the key question is whether to consider the mPE, quite often clinically silent, as the cPE. At this point, the decision to start and continue anticoagulant therapy should be based on a careful risk–benefit evaluation according to the good clinical practices.

## 5. Conclusions

During SARS-CoV-2 infection, the extent of PE is highly variable, from mPE to massive PE, sometimes also associated with arterial thrombosis. A careful clinical evaluation is needed for the proper management of anticoagulant therapy; indeed, while the use of LWMH may be preferable in the acute phase of infection, DOACs could be safely and effectively administered as clinical stability is reached. In particular, in the absence of a high bleeding risk and other contraindications to DOACs, also patients with mPE should continue anticoagulant therapy after discharge and all patients should undergo regular follow-up.

Further studies are warranted to better understand the etiopathogenesis of PE during SARS-CoV-2 infection and, consequently, the most appropriate treatment of this condition.

## Figures and Tables

**Figure 1 biomedicines-11-01282-f001:**
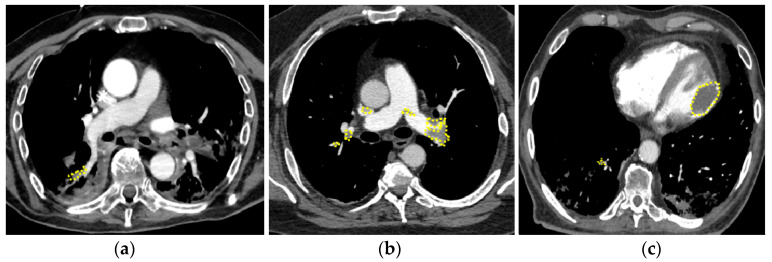
Pulmonary embolism in patients with COVID-19. (**a**) mPE with involvement of right segmental artery; (**b**) cPE with bilateral massive thrombo-embolism; concomitant left ventricle thrombosis (**c**). mPE = micro-PE, cPE = central PE.

**Figure 2 biomedicines-11-01282-f002:**
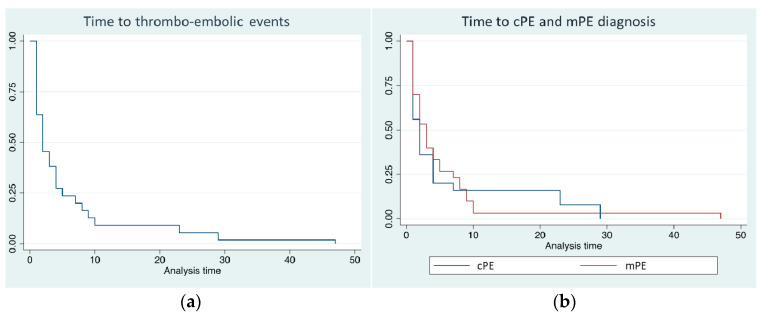
(**a**) Time to diagnosis of overall pulmonary embolism (PE) events from hospital admission; (**b**) time to diagnosis of cPE (blue line) and mPE (red line) (log-rank 0.468).

**Table 1 biomedicines-11-01282-t001:** Clinical characteristics of population and difference between cPE and mPE groups.

	All	cPE	mPE	*p* Value
Gender (male), *n* (%)	36 (72%)	18 (58%)	18 (72%)	0.42
Age (years), *mean* (±SD)	78 ± 15	76 ± 13	81 ± 16	0.02
Weight (kg), *mean* (±SD)	76 ± 19	81 ± 23	70 ± 12	0.059
BMI, *mean* (±SD)	26 ± 5	27 ± 5	25 ± 4	0.06
CHA_2_DS_2_-VASc, *mean* (±SD)	3.86 (±2.10)	3.86 ± 2.10	3.83 ± 2.23	0.23
HAS-BLED, *mean* (±SD)	1.93 (±1.15)	1.93 ± 1.15	2.0 ± 1.21	0.20
Charlson Comorbidity Index, *mean* (±SD)	6.28 ± 3.2	7.27 ± 3.66	5.64 ± 2.8	0.19
Smokers, *n* (%)	25 (44%)	13 (23%)	12 (21%)	0.88
Statin therapy, *n* (%)	19 (34%)	9 (16%)	10 (18%)	0.92
CAD, *n* (%)	21% (12)	6% (2)	40% (10)	0.002
Diabetes, *n* (%)	7% (4)	6% (2)	8% (2)	0.76
HTN, *n* (%)	68% (38)	71% (22)	64% (16)	0.78
COPD, *n* (%)	36% (20)	32% (10)	40% (10)	0.75
Cancer/leukemia, *n* (%)	14% (8)	29% (9)	4% (1)	0.03
D-Dimer (ng/mL), *median* (IQR)	1655 (889–3600)	1735 (1155–5175)	1380 (500–3600)	0.059
C-reactive protein (mg/dL), *mean* (±SD)	7 ± 6	7 ± 6	7 ± 5	0.99
Creatinine (mg/dL), *mean* (±SD)	1 ± 0.3	0.9 ± 0.3	1 ± 0.3	0.84
Clearance Cr (mL/min), *mean* (±SD)	74 ± 36	82 ± 38	66 ± 34	0.04
Diagnosis of PE (days from admission), *mean* (±SD)	6 ± 10	7 ± 12	6 ± 9	0.86

BMI = Body Mass Index; CAD = Coronary Artery Disease; HTN = Hypertension; COPD = Chronic Obstructive Pulmonary Disease; cPE = central pulmonary embolism; mPE = micro pulmonary embolism.

**Table 2 biomedicines-11-01282-t002:** Management of anticoagulant therapy and 3-month outcomes in patients with micro pulmonary thrombo-embolism (mPE) and central PE (cPE).

	All	cPE	mPE	*p* Value
LMWH (as starting anticoagulant therapy)	56 (100%)	31 (100%)	25 (100%)	n.s.
Switch from LMWH to oral anticoagulant therapy	46 (82%)	29 (94%)	17 (68%)	0.001
LMWH duration before switch (days from PE diagnosis)	16 ± 8	16 ± 9	15 ± 6	n.s.
Vitamin K antagonist *n* (%)	7 (15%)	4 (14%)	3 (18%)	n.s.
INR in patients on VKA, median (IQR)	2.3 (1.9–2.7)	2.4 (2.1–2.9)	2.3 (1.9–2.7)	n.s.
DOAC *n* (%)	39 (85%)	25 (86%)	14 (82%)	n.s.
30-day follow-up				n.s.
Exitus	4	2	2	
Recurrence of PE	0	0	0	
Bleedings (minor/major)	0	0	0	
3-month follow-up				n.s.
Exitus	0	0	0	
Recurrence of PE	0	0	0	
Bleedings (minor/major)	0	0	0	
Length of hospitalization (days)	28 ± 16	30 ± 18	25 ± 11	n.s.

LMWH = Low-molecular-weight heparin; OAC = Oral Anticoagulant; DOAC = direct oral anticoagulant; VKA = Vitamin K antagonist; n.s.: not significant.

## Data Availability

All data relevant to the study are included in the article and are available from the corresponding author upon request.

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
