# Peer review of "Efficacy and Safety of Anticoagulant Therapy in COVID-19-Related Pulmonary Embolism with Different Extension"

_biomedicines, 2023, doi:10.3390/biomedicines11051282_

Round 1

Reviewer 1 Report

No comments

Author Response

Dear reviewer, thank you very much for your authoritative judgment.

Reviewer 2 Report

Dear Authors!

You have presented quite an interesting data on COVID-19 related PE.
I have some remarks.

1. The title has to be changed to allow readers to catch quicky what is the main subject for the study. For example, "Efficacy and safety of standard anticoagulant therapy in COVID-19 related PE of different extension", or something like that.

2. Conclusions in the abstract and in the paper body has to be brief and relevant to main findings of the study. You have to emphasize two things. First is that there are two variants of PE in this cohort. Second is that standard anticoagulation is effective and safe for both.

3. I recommend to remove fig. 3. The scheme looks as a kind of recommendation now. These are just mere speculations not based even on hints from your findings. I recommend just to discuss that probably in mPE treatment may be prescribed for a shorter period of time. 

4. Please, state it clear on which anticoagulation stayed those patients who were not switched on OACs

5. Please, add the information of whether or not patients compliance to ACT was measured/registered after discharged and if not it has to be reported as a limitation of the study.

Line 112. Did you mean full dosage? This definition seems to be correct. The same is for line 162. Please use the definition “full therapeutic dose”, not “anticoagulant dose”

Lines 175-179 and Lines 180-184 are identical

English is quite fine

Author Response

Dear Reviewer, thanks for your authoritative the comments. Below we respond point by point to your valuable suggestions:

1. The title has to be changed to allow readers to catch quicky what is the main subject for the study. For example, "Efficacy and safety of standard anticoagulant therapy in COVID-19 related PE of different extension", or something like that.

As suggested we have changed the title to: "Efficacy and safety of anticoagulant therapy in COVID-19 related pulmonary embolism with different extension"

2. Conclusions in the abstract and in the paper body has to be brief and relevant to main findings of the study. You have to emphasize two things. First is that there are two variants of PE in this cohort. Second is that standard anticoagulation is effective and safe for both.

Thanks for the comment. We have reduced the conclusions in both the abstract and the body of the article and, as you suggested, we emphasized the aspects you mentioned.

3. I recommend to remove fig. 3. The scheme looks as a kind of recommendation now. These are just mere speculations not based even on hints from your findings. I recommend just to discuss that probably in mPE treatment may be prescribed for a shorter period of time.

Thanks for the comment. As you recommended we removed fig.3 and we discuss in the text that probably in mPE treatment may be prescribed for a shorter period of time. 

4. Please, state it clear on which anticoagulation stayed those patients who were not switched on OACs.

Thanks for the comment. The patients who were not switched on OACs continued LWMH for 10±5 days from discharge. As you suggested, we added the information.

5. Please, add the information of whether or not patients compliance to ACT was measured/registered after discharged and if not it has to be reported as a limitation of the study.

Thanks for the comment. There are no data on compliance with anticoagulant therapy after discharge. As you suggested, we added this information between limitations.

Line 112. Did you mean full dosage? This definition seems to be correct. The same is for line 162. Please use the definition “full therapeutic dose”, not “anticoagulant dose”

Thanks for the comment. In relation to heparin therapy, as You suggested, we use the definition “full therapeutic dose”.

Lines 175-179 and Lines 180-184 are identical

Thanks for the observation. We have corrected the repetition at lines at lines 180-184.

Reviewer 3 Report

I consider each study that is performed on the patients with pulmonary embolism as the important contribution to the knowledge in this area of medicine, because I feel a great appreciation to each physician saving the life of the patient with this diagnosis. PE is a life-threatening and severe clinical condition with possible long-lasting consequences in the form of chronic thromboembolic pulmonary disease and decreased survival of patients.

Therefore, the topic is original and surely relevant contribution in this field of clinical study.

I consider this study also as a unique contribution to the general knowledge, as there are no studies comparing the different types of PE during SARS-CoV-2 infection. Moreover, there is no unanimous consensus on anticoagulant therapy in the distal forms of PE during COVID-19, especially with regard to long-term treatment. If SARS-CoV-2 infection is considered a predisposing and triggering factor for PE, according to the recent guidelines, anticoagulant therapy should be continued for at least three months, but the key question is whether to consider the mPE, quite often clinically silent, as the cPE. At this point, the decision to start and continue anticoagulant therapy should be based on a careful risk-benefit evaluation according to the good clinical practices.

Thus, it fills a specific gap in the field, as there are still many questions regarding the management of SARS-CoV-2 infection that are needed to be addressed.

I sincerely appreciate the effort of the authors to propose new treatment approach aiming to help to manage anticoagulant treatment of PE more efficiently.

Therefore, I do not have any comments and recommend to publish the article in the current form.

Author Response

Dear reviewer,

thank you very much for your authoritative judgment.

We also really believe that pulmonary embolism, especially that related to SARS-CoV-2 infection, should be studied in depth. We hope that our study’s findings can contribute to the purpose.